# Spatiotemporal Localisation of Heparan Sulphate Proteoglycans throughout Mouse Lens Morphogenesis

**DOI:** 10.3390/cells12101364

**Published:** 2023-05-11

**Authors:** Tayler F. L. Wishart, Frank J. Lovicu

**Affiliations:** 1Molecular and Cellular Biomedicine, School of Medical Sciences, The University of Sydney, Sydney, NSW 2006, Australia; 2Save Sight Institute, The University of Sydney, Sydney, NSW 2006, Australia

**Keywords:** lens development, heparan sulphate proteoglycan (HSPG), PAPSS, syndecan, glypican

## Abstract

Heparan sulphate proteoglycans (HSPGs) consist of a core protein decorated with sulphated HS-glycosaminoglycan (GAG) chains. These negatively charged HS-GAG chains rely on the activity of PAPSS synthesising enzymes for their sulfation, which allows them to bind to and regulate the activity of many positively charged HS-binding proteins. HSPGs are found on the surfaces of cells and in the pericellular matrix, where they interact with various components of the cell microenvironment, including growth factors. By binding to and regulating ocular morphogens and growth factors, HSPGs are positioned to orchestrate growth factor-mediated signalling events that are essential for lens epithelial cell proliferation, migration, and lens fibre differentiation. Previous studies have shown that HS sulfation is essential for lens development. Moreover, each of the full-time HSPGs, differentiated by thirteen different core proteins, are differentially localised in a cell-type specific manner with regional differences in the postnatal rat lens. Here, the same thirteen HSPG-associated GAGs and core proteins as well as PAPSS2, are shown to be differentially regulated throughout murine lens development in a spatiotemporal manner. These findings suggest that HS-GAG sulfation is essential for growth factor-induced cellular processes during embryogenesis, and the unique and divergent localisation of different lens HSPG core proteins indicates that different HSPGs likely play specialized roles during lens induction and morphogenesis.

## 1. Introduction

During early embryogenesis, the morphogenesis of the eye lens begins with the proximal interaction between the surface head ectodermal cells with the evaginating optic vesicle, resulting in a thickening of this apposing ectoderm to become the lens placode. The apical constriction of these cells leads to placode invagination to form the lens pit, which will separate from the neighbouring head ectoderm to form the hollow lens vesicle. The cells at the anterior aspect of the lens vesicle will differentiate to form the lens epithelium, while the posterior lens vesicle cells elongate to form the primary lens fibre cells. The elongation of these primary fibres that extend to reach the overlying epithelium rids the vesicle of its lumen to form the basic structure of the lens [1,2]. From this point onwards, the lens continues to grow, as proliferating epithelia at the lens equator exit the cell cycle and elongate and differentiate to form secondary fibre cells, displacing the older fibre cells towards the centre of the lens [1,2]. As fibre cells mature, there is autophagy of their nuclear material and organelles, and the formation of specialized membrane interdigitations, as they acquire their distinctive transparency, a characteristic that allows the lens to perform its primary function to transmit and focus light onto the retina [3].

Lens development and maintenance are governed by the interplay of a complex network of signalling molecules within the ocular microenvironment [4]. Extensive research conducted over several decades has led to the identification of these crucial molecules and elucidation of their distinct and complementary functions in regulating specific morphogenic events and cellular behaviour [5,6]. Secreted primarily from the different ocular tissues, including the lens, retina and ciliary body, these diffusible growth factors, including members of the fibroblast growth factor (FGF) [7,8,9], Wnt [10,11,12], bone morphogenetic protein (BMP) [13,14] and Hedgehog (Hh) [15,16] families, that are found in the ocular milieu, influence lens cell behaviour (i.e., proliferation, migration, differentiation, and cell death) [4]. For the defined stages of lens morphogenesis, these factors need to be tightly controlled in a spatial and temporal manner to regulate the induction, and anteroposterior, left–right, and dorsoventral patterning of the lens [5,6]. While the various effects of growth factors on lens cell behaviour are well known, and often overlapping, the mechanisms governing the movement of these factors through the extracellular space, their precise arrival and signal transmission within target cells, remains unknown. Recent studies in mice and other animal models have highlighted the significance of heparan sulphate proteoglycans (HSPGs) localised to the cell membrane and extracellular matrix (ECM) of tissues, in facilitating the signalling and distribution of growth factors, respectively [17].

HSPGs are composed of a specific core protein structure, covalently linked to a small number of heparan sulphate (HS) glycosaminoglycan (GAG) chains. Of the thirteen full-time HSPGs, they are divided into three major families depending on this core protein structure: the transmembrane family (i.e., syndecans), glycerophosphatidylinositide (GPI)-anchored family (i.e., glypicans) and the secreted family (e.g., perlecan, agrin, and collagen XVIII) [17,18]. The syndecans and glypicans are generally localised to the cell surface, whereas perlecan, agrin, and collagen XVIII are more commonly found in the ECM and basement membranes [17].

In the synthesis of HS-GAG chains, HS sulfotransferases facilitate the transfer of sulphate ions (SO4-) from the sulphate-donor molecule, 3′-phosphoadenosine 5′-phosphosulfate (PAPS), to HS-GAGs. PAPS is a crucial and rate-limiting sulphate donor molecule that regulates all post-translational sulfation reactions in the golgi apparatus and is produced by cellular PAPS synthetases (PAPSS1/2) [19,20]. The negatively charged sulphate groups of the HS-GAG chains interact with the positively charged lysine/arginine-rich regions of many proteins [17], allowing HSPGs to interact with and modulate the activity of various growth factors including FGFs [21,22], BMPs [23,24], Wnts [25,26], Hh [27,28], and ECM molecules, such as integrins [29,30], laminin [31], and fibronectin [32].

The downstream effects of HSPG–protein interactions include the activation of high-affinity receptor signalling complexes, and the mediation of intracellular signalling cascades leading to the activation of key effector genes and changes in the cell behaviour and phenotype [17,33]. The pleiotropic modulation of growth factor receptor signalling by HSPGs (altering ligand density and activity) can locally stabilise mitogen and morphogen gradients at the tissue level [33]. These gradients are critical for organogenesis and for the maintenance of normal tissue homeostasis [34]. As the activity of HSPGs primarily depends on its fine structure (i.e., sulphation) and location (i.e., the cell surface or ECM) of their HS-GAG glycosaminoglycan chains, the specific activity of HSPGs in a given cell-type or tissue, at a given time can be determined by its HSPG profile: the localisation of HSPG core proteins and its complement of HS-GAG sulfation enzymes [17,33]. Previously, we have shown all HSPG core proteins are present in the postnatal rat lens, localised to its epithelia, fibres and basement membrane, the lens capsule, in unique and distinct expression patterns that align with key functional regions [35]. The distinct spatial and temporal expression of HSPGs implies that differentiating lens cells may acquire a unique set of HSPGs, leading to specific patterns of HS–protein interactions and function during lens morphogenesis. Here, we report for the first time the comprehensive spatial and temporal localisation of all HSPGs in the developing murine lens. Taken together with the current literature, we highlight putative roles for specific HSPGs during key stages of lens development, as well as provide support for the requirement of HS sulphation in growth factor-induced lens cellular processes.

## 2. Materials and Methods

### 2.1. Animals

Animal experimentation protocols were authorized by The University of Sydney Animal Ethics Committee and were conducted in accordance with the guidelines established by the National Health and Medical Research Council (Australia) as well as the Association for Research in Vision and Ophthalmology’s declaration for the Use of Animals in Ophthalmic and Vision Research. Mouse conception was recorded as midnight on the day of mating and embryos were routinely collected post conception at embryonic (E) day 9.5 (E9.5), E10.5, E11.5, E14.5, and E16.5, representing key stages of lens development (see Figure 1). Pregnant FvB/N mice were sacrificed by asphyxiation with carbon dioxide, and embryos were extracted and sacrificed by decapitation. Embryonic heads were collected for histological processing. Ten-day-old (P10) albino Wistar rats (*Rattus norvegicus*) were sacrificed by asphyxiation with carbon dioxide followed by cervical dislocation.

### 2.2. Immunolabelling of Heparan Sulphate Proteoglycans

Intact embryonic heads were collected and fixed for 24 h in 10% neutral buffered formalin (NBF; Sigma, Castle Hill, NSW, Australia) before processing for routine paraffin embedding. Paraffin-embedded tissues were serially sectioned at 5–6 μm. Mid-sagittal sections of embryonic heads from at least three independent animals were used for immunolabelling and for periodic acid–Schiff’s (PAS) staining to highlight lens morphology. Monoclonal antibodies were used to detect all highly sulphated HSPGs as previously described [35]. Embryonic head sections were first labelled for highly sulphated forms of HS-GAG (HS-GAG F58-10E4, 370255; Amsbio LLC, Abingdon, UK). We also labelled for sulphated chondroitin sulphate (CS)-GAGs (CS-56, C8035; Sigma), as some HSPGs (syndecans -1/-3, agrin and collagen XVIII) carry CS-GAG chains alongside their predominant HS-GAG chains. To characterize the spatial localisation of the different HSPGs in the developing murine lens, we immunolabelled mid-sagittal sections using commercially available antibodies (see Appendix A). For all immunofluorescent labelling studies, cell nuclei were counter-labelled with bisbenzimide (Hoechst 33258; Sigma). Images in figures are oriented with the ventral side positioned to the right and the dorsal side to the left. Controls were included to account for any effects of autofluorescence (unlabelled tissue controls), secondary antibodies (chain-specific isotype and secondary antibody alone controls), co-labelling effects (single label controls), and nuclear staining (no Hoechst controls). Tissue samples from at least three individual animals (up to 6 eyes) at different stages of development were used for each label/antibody investigated.

A light microscope equipped with epifluorescence (Leica DMLB 100S, DFC-450C camera and LAS software v4.8.0, Leica Microsystems, Wetzlar, Germany) was used for PAS-staining (brightfield). For immunolabelling (immunofluorescence) studies, images were captured using a ZEISS LSM-800 microscope with an AxioCam 506 camera and ZEN Blue Edition v2.3 software (Carl Zeiss, Jena, Germany), or a ZEISS AxioScan.Z1 microscope with a Hamamatsu ORCA-flash 4.0 v2 camera and ZEN SlideScan 2012 software (Carl Zeiss). All images were saved as uncompressed tagged information field format (TIFF) files at 16-bit depth. Image processing was performed using Photoshop v21.1.1 (Adobe, Inc., San Jose, CA, USA) and ImageJ (FIJI) v2.0 (Wayne Rasband, NIH, Bethesda, MD, USA) on the Mac OS X 10.15.6 (Apple Inc., Cupertino, CA, USA) and Windows 10 (Microsoft Inc., Seattle, WA, USA) operating systems.

## 3. Results

A summary of the labelling for HS-GAG, CS-GAG, PAPSS2, and all thirteen HSPG core proteins during murine lens development is outlined in Table 1.

### 3.1. PAPS Synthetase-2 Is Localised to the Developing Mouse Lens

We first immunolabelled the PAPSS2 (HSPG sulfation) in the developing mouse lens (Figure 2). PAPSS2 reactivity was absent from all early eye structures, including cells comprising the lens placode (E9.5, Figure 2(A1)), and lens pit (E10.5, Figure 2(A2)). It was not until E11.5, in the lens vesicle that PAPSS2 immunolabelling appeared, strongly localised to the elongating primary lens fibre cells (Figure 2(A3)). This fibre-specific label persisted and was also observed in the early secondary lens fibre cells at E14.5 (Figure 2(A4–A7)) and E16.5 (Figure 2(A8–A10)). At E16.5, PAPSS2 reactivity was also evident in the anterior lens epithelium but was not present in the lens capsule (Figure 2(A9)).

### 3.2. Heparan Sulphate Is the Predominant Sulphated Glycosaminoglycan Expressed throughout Murine Lens Development

HSPG core proteins can carry HS-GAG GAG chains or a combination of both HS-GAG and CS-GAG chains. To determine which of these two GAG species represents the major “active” sulphated fraction of HSPGs in the developing lens, the distribution of HS-GAG and CS-GAG was compared during key stages of lens development (Figure 3(A1–A8,B1–B8)) using antibodies specific for the sulphated domains on HS-GAG (HS-10E4, Figure 3(C1–C8)) or CS-GAG (CS-56, Figure 3(D1–D8)) chains. HS-GAG chain labelling was first observed at very low levels in the cells of the lens placode (E9.5, Figure 3(C1)) with stronger labelling in the lens pit (E10.5, Figure 3(C2)). At E11.5 (Figure 3(C3)), HS-GAG labelling was prominent in the developing lens capsule, anterior lens epithelium and ubiquitously in elongating primary lens fibre cells. HS-GAG reactivity was maintained in the lens capsule and cytoplasm of anterior epithelial cells at E14.5 (Figure 3(C4)) and E16.5 (Figure 3(C5–C8)). The HS-GAG chain labelling was the strongest throughout the cytoplasm of secondary lens fibre cells (Figure 3(C4)), with decreased reactivity at the germinative and transitional zones (Figure 3(C4,C8)). The concentrated immunolabelling of HS-GAG was also observed at the posterior tips (E14.5, Figure 3(C4)) and in the nuclei (Figure 3(C4,C7)) of elongating secondary fibre cells. Although absent from the extracellular matrix of the early developing eye (E9.5 and E10.5; Figure 3(B1,B2)), HS-GAG was strongly associated with the matrix connecting the posterior lens vesicle and early retina (Figure 3(C3)), and the developing vitreous humour from E14.5 (Figure 3(C4)). In contrast to the localisation of active HS-GAG chains in early lens development, CS-GAG chain labelling was primarily observed in the pericellular matrix (Figure 3(D1–D3)). CS-GAG immunolabelling was completely absent from the lens placode (Figure 3(D1)), lens pit (Figure 3(D2)), and lens vesicle (E11.5, Figure 3(D3)). Similarly, CS-GAG reactivity was first observed at low levels in the E14.5 lens capsule, anterior lens epithelium, and anterior pole of the secondary lens fibre cells (Figure 3(D4)). CS-GAG immunostaining was strongly correlated with the extracellular matrix at all early stages of eye development, particularly in the preplacode matrix (E9.5; Figure 3(D1)), and the interstitial matrix connecting the invaginating lens pit and optic cup (E10.5; Figure 3(D2)) and surrounding the lens vesicle (E11.5; Figure 3(D3)). CS-GAG was weakly evident in the ocular humours in the later stages of eye development (E14.5 and E16.5; Figure 3(D4–D6)). CS-GAG immunolabelling was absent from the lens capsule at E16.5 (Figure 3(D5,D6,D8)), and decreased in the fibre cells relative to E14.5 (Figure 3(D7,D8)). The CS-GAG labelling remained in the anterior lens epithelium (Figure 3(D5)) and decreased in the germinative and transitional zones (Figure 3(D8)).

### 3.3. Syndecan HSPGs Are Differentially Spatially Localised in the Lens throughout Development

The different syndecan HSPGs differ in their expression, both temporally and spatially, at all major stages of eye development. Syndecan-1 remains absent from the developing lens until E16.5, where it is strongly localised to the lens capsule and posterior vessels of the tunica vasculosa lentis (tvl). Weak syndecan-1 staining is also observed in a subset of anterior lens epithelial cells and anterior secondary fibre cell tips (Figure 4(A5)). Syndecan-2 (Figure 4(B1)), -3 (Figure 4(C1)) and -4 (Figure 4(D1)) are found at E9.5 in both the optic vesicle and the lens placode. The expressions of these syndecans persist at embryonic day 10.5, with strong labelling in the lens pit and optic cup (Figure 4(B2,C2,D2)). Syndecan-2 strongly stained throughout the invaginating lens pit, while syndecan-3 and -4 labelled particularly strong in the posterior cells of the lens pit (Figure 4(C2,D2)). At E11.5, the prospective lens epithelial cells and elongating primary lens fibre cells showed strong labelling of syndecan-2 (Figure 4(B3)) and -3 (Figure 4(C3)). For syndecan-2, this label was restricted to the primary lens fibres and absent from the anterior aspect of the lens (Figure 4(D3)). Syndecan-1 was absent from the lens up to E11.5, where it first appears to localise to the inner aspect of the retina (Figure 4(A3)). At E14.5, the localisation of each of the syndecans was distinct from each other. Syndecan-1 was primarily localised to the differentiating retinal ganglion cells, and the lens capsule and blood vessels (Figure 4(A4)). Syndecan-2 was strongly localised to the lens capsule, tvl, lens epithelial cells, and nuclei of fibre cells. It also showed strong localisation in the cornea and developing iris (Figure 4(B4)). Syndecan-3 (Figure 4(D3)) and -4 (Figure 4(D4)) were both present in the lens epithelium and fibres in different patterns. Syndecan-3 staining was primarily observed in the secondary lens fibres, concentrated on the anterior fibre tips. It was absent from newer secondary lens fibre cells (Figure 4(C4)). On the other hand, syndecan-4 staining was primarily observed in early secondary fibre cells, with distinct staining at the anterior and posterior fibre cell tips (Figure 4(D4)). These expression patterns were maintained at E16.5 for syndecan-2 (Figure 4(B5–B8)), -3 (Figure 4(C5–C8)), and -4 (Figure 4(D5–D8)). At E16.5, syndecan-1 was still absent from the lens cells, but strongly localised to the lens capsule (Figure 4(A5–A8)).

### 3.4. Glypican HSPGs Differ in Temporal Expression at Key Stages of Lens/Eye Development

At E9.5, glypican-4 is exclusively localised to the pre-placodal matrix (Figure 5(D1)), while glypican-2, -5, and -6 are present in both the lens placode cells and optic vesicle (Figure 5(B1,C1,E1,F1)). Glypican-1 was absent at this early stage (Figure 5(A1)). By E10.5, all glypicans were present in the lens pit and optic cup, apart from glypican-4, that was localised to the extracellular matrix between the lens pit and optic cup (Figure 5(D2)). The label for glypican-3 and -5 was concentrated upon the basal aspect of the lens pit in the presumptive primary lens fibre cells (Figure 5(C2,E2)). Glypican-1, -2, -5, and -6 were also present in the surface head ectoderm (Figure 5(A2,B2,E2,F2)). All glypicans were present in the lens vesicle (Figure 5(A2,B2,C2,E2,F2)), except for glypican-4, that was primarily localised to the lens capsule, differentiating retinal ganglion cells, and posterior lens blood vessels (Figure 5(D3)). At E11.5, glypican-3 stained all lens vesicle cells diffusely, with more intense staining in the presumptive lens epithelial cells (Figure 5(C3)). Glypican-1, -2, -5, and -6 were predominantly localised to the posterior elongating primary lens fibre cells (Figure 5(A3,B3,E3,F3)). These glypicans were mostly localised in a concentrated region at the posterior tips of the primary lens fibres, apart from glypican-6, that strongly labelled all fibre cells (Figure 5(F3)). From E14.5, the expression of glypicans was generally consistent. Glypican-1 (Figure 5(A4–A8)), -2 (Figure 5(B4–B8)), -3 (Figure 5(C4–C8)), and -5 (Figure 5(E4–E8)) were predominantly localised to the lens epithelium and newly differentiating secondary lens fibre cells, where perinuclear staining was observed for all four glypicans (Figure 5(A7,B7,C7,E7)). Glypican-4 was absent from lens cells, with some punctate staining in the anterior lens epithelium, and strong localisation to the lens capsule (Figure 5(D4–D8)). Glypican-6 was absent from the epithelium and transitional zone, but strongly labelled the cytoplasm of the fibre cells (Figure 5(F4–F8)).

Postnatally, in the 10-day-old rat lens, labelling for glypican-5 was localised to both epithelial and fibre cells, with stronger labelling at the germinative and transitional zones (Appendix A). Glypican-5 was also found in the corneal epithelium and endothelium (Appendix A), ciliary non-pigmented epithelium (Appendix A), and in remnant blood vessels of the tunica vasculosa lentis (Appendix A). Glypican-5 was absent from the lens capsule (Appendix A).

### 3.5. Localisation of High-Molecular-Weight HSPGs Extends beyond the Lens Capsule

Immunoreactivity for perlecan is weak in early lens structures (Figure 6(A1,A2)) but labelling appears to be stronger in the matrix and basement membranes of the E11.5 eye (Figure 6(A3)). At this stage, perlecan is observed in the anterior cells of the lens vesicle (Figure 6(A3)). This pattern of staining is maintained in the fully formed lens at E14.5 and E16.5, with perlecan mostly restricted to the lens capsule and anterior lens epithelium (Figure 6(A4–A8)). Diffuse perlecan staining was observed in the nuclei of some differentiating lens fibre cells (Figure 6(A7)). Both collagen XVIII/endostatin (Figure 6(B1,B2)) and agrin (Figure 6(C1,C2)) were observed in the cells of the lens placode and lens pit. Collagen XVIII/endostatin reactivity was restricted to a punctate label in the primary lens fibres of the lens vesicle (Figure 6(B3)). This label persisted in the E14.5 lens (Figure 6(B4)). At E16.5, collagen XVIII/endostatin labelled in condensed patches throughout the cytoplasm of both lens epithelial and fibre cells (Figure 6(B5–B8)). On the other hand, agrin stained strongly throughout the cells of the lens vesicle, concentrated apically in the presumptive anterior lens epithelial cells, and basally in the new elongating primary lens fibre cells (Figure 6(C3)). From E14.5, agrin immunoreactivity was strongly visible throughput the cytoplasm of the central lens fibre cells (Figure 6(C4)), and this label persisted in the E16.5 lens (Figure 6(C5–C8)). Agrin was clearly associated with a subset of fibre cell nuclei (Figure 6(C4,C7)). Agrin staining was almost completely absent from the anterior lens epithelium and transitional zone (Figure 6(C4,C5,C8) and Table 1).

**Table 1 cells-12-01364-t001:** Summary of HSPG immunolocalisation in mouse eye lens development.

Embryonic Day		GAG	Other	Core Proteins
**E9.5**	Lens placode	HS		Sdc2, Sdc3, Sdc4, Gpc2, Gpc4, Gpc5, Gpc6, perlecan, collagen XVIII, agrin
	Optic vesicle	HS, CS		Sdc2, Sdc3, Sdc4, Gpc2, Gpc5, Gpc6, perlecan, collagen XVIII, agrin
	Preplacodal matrix	CS		Sdc2, Sdc4, Gpc4, Agrin
**E10.5**	Lens pit	HS		Sdc2, Sdc3, Sdc4, Gpc1, Gpc2, Gpc3, Gpc5, Gpc6, perlecan, collagen XVIII, agrin
	Head ectoderm	HS		Sdc2, Sdc3, Sdc4, Gpc1, Gpc2, Gpc3, Gpc4, Gpc5, Gpc6, perlecan, agrin
	Optic cup	HS, CS		Sdc2, Sdc3, Sdc4, Gpc1, Gpc2, Gpc3, Gpc4, Gpc5, Gpc6, perlecan, collagen XVIII, agrin
	Extracellular matrix (lens pit-optic cup)	CS		Sdc2, Gpc4, perlecan?
**E11.5**	Anterior lens epithelium	HS	PAPSS2	Sdc2, Sdc3, Sdc4, Gpc1, Gpc2, Gpc3, perlecan, collagen XVIII, agrin
	1° lens fibre cells	HS	PAPSS2	Sdc2, Sdc3, Sdc4, Gpc1, Gpc2, Gpc3, Gpc5, Gpc6, perlecan, collagen XVIII, agrin
	Lens capsule	HS		Sdc1, Sdc2, Gpc4, perlecan
**E14.5**	Anterior lens epithelium	HS *, CS		Sdc2 *, Sdc3, Sdc4, Gpc1, Gpc2, Gpc3, Gpc4, perlecan, collagen XVIII
	2° lens fibre cells	HS *, CS	PAPSS2	Sdc2 *, Sdc3, Sdc4, Gpc1, Gpc2, Gpc3, Gpc5, Gpc6, collagen XVIII, agrin
	Lens capsule	HS, CS		Sdc1, Sdc2, Gpc2, Gpc4, perlecan
	Tunica vasculosa lentis	HS, CS		Sdc1, Sdc2, Gpc1, Gpc2, Gpc4, Gpc5, perlecan
**E16.5**	Anterior lens epithelium	HS *, CS		Sdc2 *, Sdc3, Sdc4, Gpc1 *, Gpc2 *, Gpc3 *, Gpc5, perlecan, collagen XVIII, agrin
	2° lens fibre cells	HS *, CS	PAPSS2	Sdc2 *, Sdc3, Sdc4, Gpc1 *, Gpc2 *, Gpc3 *, Gpc5, Gpc6, perlecan *, collagen XVIII, agrin *
	Lens capsule	HS		Sdc1, Sdc2, Gpc4, perlecan
	Tunica vasculosa lentis	HS, CS		Sdc1, Sdc2, Gpc1, Gpc2, Gpc4, Gpc5, perlecan

* Nuclear localisation in cells.

## 4. Discussion

While there are many reports to date studying vertebrate lens morphogenesis and the growth factors involved, less is understood about how these different growth factors are positioned to drive this development. Given the important role of HSPGs in regulating many growth factors, together with recent studies implicating HSPGs as essential for the induction and development of the eye, here we present the most comprehensive spatiotemporal labelling of all the different HSPG core proteins and their associated GAGs throughout murine lens development. This study complements and extends from our earlier report on the localisation of HSPGs in the postnatal rat lens [35], wherein we described how the defined patterns of all HSPGs associate with lens cellular activity. Of note is that our earlier study reported the absence of glypican 5 in the postnatal lens, of which we now have corrected and clearly show that it is indeed also present in the lens when using a more selective and effective commercial antibody. Given that all HSPGs are expressed in the lens, by linking the defined localisation of their respective HSPG core proteins, along with their reported mode of activity, it can provide valuable insights into their functional roles [36,37].

The activity of HSPGs is largely dependent on the distinct sulfotransferases of their HS chains, which source sulphate from PAPS produced by PAPSS1/2 synthetases [38,39]. Transcriptome data identify both *papss1* and *papss2* in the lens [40], with strong PAPSS2 protein labelling in the developing murine and postnatal rat eye, specifically in the lens [38]. Here, we found that PAPSS2 was absent in very early lens induction (i.e., lens placode, lens pit), with strong reactivity first appearing in the lens vesicle, distinctly labelling the elongating primary lens fibre cells. This fibre-specific label persisted and was later observed in secondary lens fibres at E14.5 and E16.5 when the differentiating lens epithelium also began to label strongly for PAPSS2. The weak to no immunolabelling for PAPSS2 in the lens placode and pit is of interest as we see the strong labelling of not only HS-GAG but many HSPGs that require PAPSS for their sulfation. The absence of PAPPS2 suggests that the activity of HSPGs in early lens induction may be largely mediated by HSPG core protein–protein interactions, and is independent of GAG–protein interactions, implying an expected low sulfation and binding potential of these HSPGs with ocular growth factors. An example of this is glypican-4, that only carries HS chains but does not overlap in expression with HS-GAG. Alternatively, it is also plausible that the other PAPSS isoform, PAPSS1 (encoded by the *papss1* gene), is the predominant sulphate donor at this early stage of lens development, consistent with transcript expression profiles from published embryonic mouse lens datasets [41,42,43,44].

The selectivity of interactions between HSPGs and proteins is not solely determined by the type of core protein (such as syndecan or glypican) or its location (for example, on the cell surface or in the extracellular matrix), but also by the precise structure (i.e., sulphation) of the HS-GAG chains. Although HSPGs are primarily composed of HS-GAGs, certain species such as syndecan-1/-3 [45,46], agrin [47], and collagen XVIII [48] may also carry CS-GAGs. Lens cells produce some proteoglycans carrying both HS-GAG and CS-GAG, although HS-GAG is the most common [35,49,50]. Consistent with this, here we report that HS-GAG, and not CS-GAG, is the predominant GAG associated with HSPGs throughout murine lens development. Unlike HS-GAG, CS-GAG was absent from the lens until E14.5, indicating that HS-GAGs are likely the major species regulating the early events of lens induction, invagination, and cell fate specification. The presence of CS-GAG in the pre-placodal and peri-lenticular matrix throughout development may suggest a role in cell–matrix interactions or morphogen gradient formation**. In** contrast, HS-GAG was absent from the ocular matrix throughout the early stages of lens development, instead appearing and localising predominantly to the presumptive and differentiating lens cells. These patterns were often associated with regions of lens cell activity (i.e., lens placode, posterior aspect of lens pit/vesicle and lens transitional zone). The expression patterns of HS-GAG and CS-GAG in the developing lens suggest that both GAG types may have overlapping and distinct roles in regulating functional cellular processes. For instance, fibre differentiation may be regulated by both GAG types, while morphogen gradient formation in the lens capsule may be predominantly regulated by HS-GAG. In other tissues, such as the developing brain, HS-GAG and CS-GAG have been shown to serve distinct functional roles, with CS-GAG stabilizing existing synaptic connections and HS-GAG stimulating the formation of new synapses [51]. These functional differences are likely due to the selective interaction of each GAG type with different proteins in a structure-dependent manner [52]. The reported differential localisation of HS-GAG and CS-GAG [51,53,54] is consistent with their distinct functional roles [55].

Most of the HSPG core proteins that we examined localised to the cells of the surface head ectoderm and optic vesicle; however, syndecan-1 and glypican-1 were both absent at this stage. Despite being a traditionally cell-associated HSPG, glypican-4 was absent from the head ectoderm and optic vesicle but was strongly localised to the pre-placodal matrix. Similarly, syndecan-2, which is traditionally a cell-associated HSPG, was also observed in the pre-placodal matrix, along with the surface head ectoderm and optic vesicle, consistent with previous reports of syndecan-2 being a matrix-associated HSPG in ocular tissues [35,56]. The pre-placodal matrix is thought to prevent this defined region of proliferative head ectodermal cells from spreading, so that their continued proliferation and progressive cell crowding thickens this region of the ectoderm to form the lens placode [57]. The regulated signalling activities of several mitogens and morphogens, such as BMPs, Hh, FGFs, and Wnts, orchestrate the inductive events associated with the optic vesicle-presumptive lens interaction [58,59], and all these growth factors have been shown to be regulated by the HSPGs that we localise to the lens placode [60,61,62,63,64]. FGF, Hh, BMP, and Wnt-signalling are each essential regulators in eye development, with their on/off switching essential for different stages of lens development [6], and defects in their signalling activity shown to compromise lens development. While the sustained inductive signalling of BMP and FGF are essential for lens placode development and the early stages of lens specification [65], these early stages also require the inhibition of Hh- [16] and Wnt- [66] signalling, with heightened Hh-signalling activity shown to suppress lens formation in zebrafish [67]. Later in lens morphogenesis, Wnt-signalling is activated to promote epithelial cell adhesion, integrity, and polarity, as well as fibre differentiation [11,66,68,69]. Constitutively active Hh-signalling through its receptor, patched, promotes epithelial maintenance at the expense of fibre differentiation [70]. Thus, the inhibition of Hh-signalling, once again with fibre cell differentiation, is essential for normal lens development. BMP-signalling promotes apical constriction and pro-survival signals during lens placode invagination and drives proliferation of lens epithelial cells and cell cycle exit at the transitional zone, that is bolstered by FGF-signalling driving lens epithelial cell proliferation and fibre differentiation [13,14,71].

The spatiotemporal regulation of the ocular morphogens, together with other extracellular signalling proteins, forms the basis of lens patterning, polarity, growth, and cell fate specification [1,6,72], that are regulated by the many HSPGs we report to be expressed herein. Many of these same HSPGs regulate other matrix-related proteins, including β1-integrins and fibronectin [73,74,75], that are also co-localised to the early lens placode cells, and play an essential role in this inductive process [76,77,78,79].

Glypicans are predominantly expressed during embryonic development, and increasing evidence suggests that they are major regulators of morphogenic signalling and gradient events in development, controlling left/right and dorsal/ventral patterning, as well as controlling tissue size and growth cues [80]. Genetic and biochemical studies have shown that glypicans can stimulate or inhibit the signalling pathways triggered by Wnts, Hh, BMPs, and FGFs, that have been mentioned to be responsible for patterning and tissue formation during embryogenesis, depending on the context of the tissue to drive growth patterning events [80]. Not surprisingly, mutations in genes encoding glypicans often result in developmental abnormalities or overgrowth [80]. For instance, Glypican-3 has been shown to inhibit hedgehog signalling during development by competitively binding to the ligand, preventing it from binding to patched, through a HS-GAG independent process [61]. Similarly, Glypicans-1, -3, and -4 negatively regulate BMP-signalling by competing with BMP receptors for ligand binding in a similar fashion [81,82]. On the other hand, glypicans seem to both positively and negatively regulate Wnt-signalling in different developmental systems. Glypican-4 promotes cardiac specification and differentiation by attenuating canonical Wnt- and BMP-signalling [81], while glypican-1 can act as a Wnt-coreceptor to activate Wnt-signalling [82]. For different morphogens, glypicans seem to mainly activate or enhance signalling. Glypicans can interact with FGFs and their receptors to stabilize their assembly and enhance downstream ERK1/2 or PI3K/Akt, as well as bind to TGF-β and its receptors to promote SMAD-signalling [82].

As the mouse lens placode and optic vesicle invaginate at E10.5, we find that most of the HSPG core proteins localise to the cells of the lens pit, with syndecan-1 and glypican-4 yet to appear. Glypican-4 was predominantly localised to the surrounding matrix together with syndecan-2 and perlecan. Here, we begin to observe subtle changes between some of the core proteins, such as syndecan-4, that appears to distinctly label the posterior tips of lens pit cells, with the majority of the other core proteins localising diffusely throughout all the lens pit cells. BMP-4/-7 activity [83] and the downstream activation of GTPases (RhoA, Rac1, ROCK) drive changes in cytoskeletal architecture essential for cell elongation, apical constriction, and epithelial invagination during lens pit formation [84]. Glypican-3 is well established as a primary regulator of cellular responses of BMP-4/-7 in developing tissues [85], although emerging evidence suggests that other HSPGs, such as agrin [86] and syndecan-3 [87], may also play tissue-specific roles in BMP regulation. On the other hand, the syndecans are major regulators of RhoA/Rac1/ROCK GTPase-signalling and cytoskeletal modifications, through interactions with the intracellular domains of their core proteins [88,89,90].

As the mouse lens acquires more of an antero-posterior polarity with the formation of the lens vesicle at E11.5, we see that the acute change in posterior lens vesicle cell elongation to form the first primary fibres is associated with much HSPG activity. In the first instance, we not only have PAPPS2 appearing in these cells, but also co-localising with HS-GAG and the core proteins of most HSPGs, suggesting their putative roles in the differentiation, attachment, and elongation of primary lens fibre cells. Syndecan-1 continues to remain absent not only from lens cells but from all developing eye structures. Given that PAPSS2 appears distinctly at this developmental stage, overlapping with many HSPGs and HS-GAGs, this suggests a marked rise in HS sulfotransferase activity that may be essential to drive primary lens fibre differentiation. With the differentiation of the primary fibres, we also have the concomitant differentiation of the anterior lens vesicle cells to form the epithelium. Lens epithelial cell differentiation is known to be regulated by Wnt/beta-catenin signalling [68] which in turn is regulated by glypican-3 [91], that is perfectly positioned for this, given the strong reactivity we observe in the anterior lens cells.

While we show that glypicans appear to generally localise in similar patterns to syndecans, the glypicans display temporal differences in their onset. This may be indicative of different core proteins regulating different developmental/cell fate specification/patterning events. It may be the case that the variable spatiotemporal localisation of the different HSPGs, particularly glypicans, throughout lens development is representative at least in part by their differential regulation of the morphogenetic switches in signalling events [80]. Although no lens-specific phenotypes have been reported for animals deficient in glypicans, compensatory regulation between closely related HSPGs (i.e., glypicans-2/-4 and -3/-5) is likely. Our understanding of this family of HSPGs and how they function in different tissues is still very limited.

The larger secreted HSPGs, perlecan [35,92,93,94], collagen XVIII [35,92], and agrin [35,95] are recognized as the predominant HSPGs in the postnatal lens capsule; however, we have previously reported different patterns of other HSPG core proteins (syndecans-1/-2/-4 and glypicans-2/-4) in the postnatal rat lens capsule: syndecan-2 and perlecan are abundant throughout the lens capsule, while syndecan-4, glypican-2, glypican-4, collagen XVIII/endostatin, and agrin are restricted to the outer reticular lamina of the lens capsule [35]. In contrast, syndecan-1-labelling appeared to be uniquely limited to the basal lamina. In the developing mouse eye, we only observed a subset of these HSPGs in the lens capsule (syndecans-1/-2/-4, glypican-4 and perlecan), suggesting that some HSPGs (i.e., glypican-2, collagen XVIII, and agrin) are deposited in the capsule at later stages. Of the minor HSPG core proteins we observed in the developing lens, three are syndecans, that bind to several structural and matrix regulatory proteins found in the lens capsule (e.g., collagen IV, laminin, fibronectin and integrins), and have previously been implicated as key regulators of ECM assembly, cell–matrix interactions, and cell adhesion [38].

Perlecan exists in an antero-posterior gradient in the lens capsule, consistent with complimentary antero-posterior gradients of FGF bioavailability and signalling activity [96,97]. Perlecan HSPGs are also required for the structural integrity of the lens capsule [36], as it is shown here to be progressively deposited to the capsule over the course of lens development, consistent with its label in the embryonic chick lens [98]. While the other large secreted high-molecular-weight HSPGs, collagen XVIII, and agrin are often reported to be associated with ECM and basement membranes, here we report that both these core proteins are present from the earliest stages of lens development, primarily within the cells of the presumptive and developing lens, and are not observed in the lens capsule, as we previously reported in the postnatal rat lens [38]. Rather, they are primarily associated with the lens epithelium, and show particularly strong staining in lens fibre cells. Mice deficient for *col18a1*, encoding the collagen XVII core protein, have a widespread malformation of ocular tissues [99,100], with the overexpression of the endostatin domain of Collagen XVIII leading to cataract and structurally compromised basement membranes [101,102]. The overexpression of agrin also produces a dysgenic phenotype that is specific to the eye [95]. Similarly, the knock-down of agrin in zebrafish embryos results in a small eye and lens phenotype, leading to decreased HS-GAG staining in the lens, but not retina, suggesting that other HSPGs may be upregulated in response to this loss of agrin [103]. The knockdown of agrin in zebrafish also leads to selectively decreased FGF-signalling and the downstream activity of MAPK/ERK in the lens, providing strong evidence that agrin is likely regulating FGF-ERK signalling activity in the developing lens [103]. Interestingly, the strong pattern of HSPG-labelling in the posterior lens vesicle cells mirrors that of phospho-ERK1/2 staining [104]. Since FGF/ERK-signalling is required for lens fibre cell elongation [105], and agrin is strongly localised to both the rapidly elongating primary and secondary lens fibres, it may be playing an important role in FGF-induced lens fibre differentiation. Notably, agrin is absent from the transitional zone of the lens, where cells exit the cell cycle and begin to differentiate into fibres, instead of selectively staining the elongated/ing lens fibres.

Here, we report on the immunolabelling of HS-GAG and specific core proteins in the nuclei of lens epithelial cells (syndecan-2) and fibre cells (syndecan-2, glypican-1, glypican-5, perlecan, and agrin). Some of these HSPGs have previously been reported in the cell nuclei of other tissues, such as glypican-1 (neurons, glial cells); [106] and agrin (developing rat brain cells) [107], while other HSPGs (i.e., glypican-5), to the best of our knowledge, are reported herein for the first time in the nuclei of lens cells. HS-GAG and HSPG core proteins in select cell types have been proposed to play roles in the transport of cargo or growth factor/receptor complexes to the cell nucleus (e.g., FGF/FGFR) to regulate gene expression [35,108]. The presence or absence of HSPGs correlates with cellular function, phenotype, and changes in cell cycle phases, suggesting that their placement in the nucleus may control cellular function [35,108]. Here, we report that most HSPGs associated with lens cell nuclei are present in the transitional zone or in new fibre cells, where the lens epithelial cells exit the cell cycle and undergo major morphological changes to differentiate into fibre cells, suggesting that these specific HSPGs may play a regulatory role in these processes. Further investigations are necessary to elucidate the specific functional roles of HSPGs in lens cells. The current study is the first to comprehensively map all HSPG localisation in a developing tissue during organogenesis. The unique patterns of expression observed here for HSPG core proteins in the developing lens imply that, during differentiation, lens cells acquire a distinct set of HSPGs that may interact selectively with proteins, resulting in distinct patterns of HS-protein binding and signalling regulation. Moreover, our findings suggest that different HSPGs may play distinct and overlapping roles at different stages of lens induction and morphogenesis. In conjunction with earlier research, our findings also suggest that HS-GAG sulfation is necessary for growth factor-induced cellular processes in the lens, and that PAPSS2 plays a distinct temporal role as a sulphate donor. By further characterizing and specifically modulating the different sulphated HSPGs expressed in the lens, we hope to better understand how HSPGs regulate different growth factor-mediated signalling, and its respective lens cell behaviour. With this, the specific roles of HSPGs in modulating growth factor activity could facilitate the development of innovative approaches for manipulating cellular behaviour. Such strategies could have broader implications beyond the context of lens biology and diseases, extending to other growth factor-mediated pathologies.

## Figures and Tables

**Figure 1 cells-12-01364-f001:**
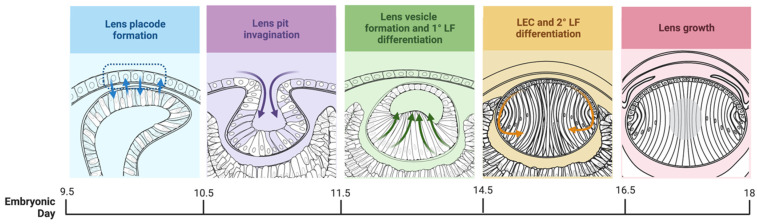
Representative stages of murine lens development. Arrows in lens placode formation indicate inductive interactions. Other arrows indicate cell movements/growth.

**Figure 2 cells-12-01364-f002:**
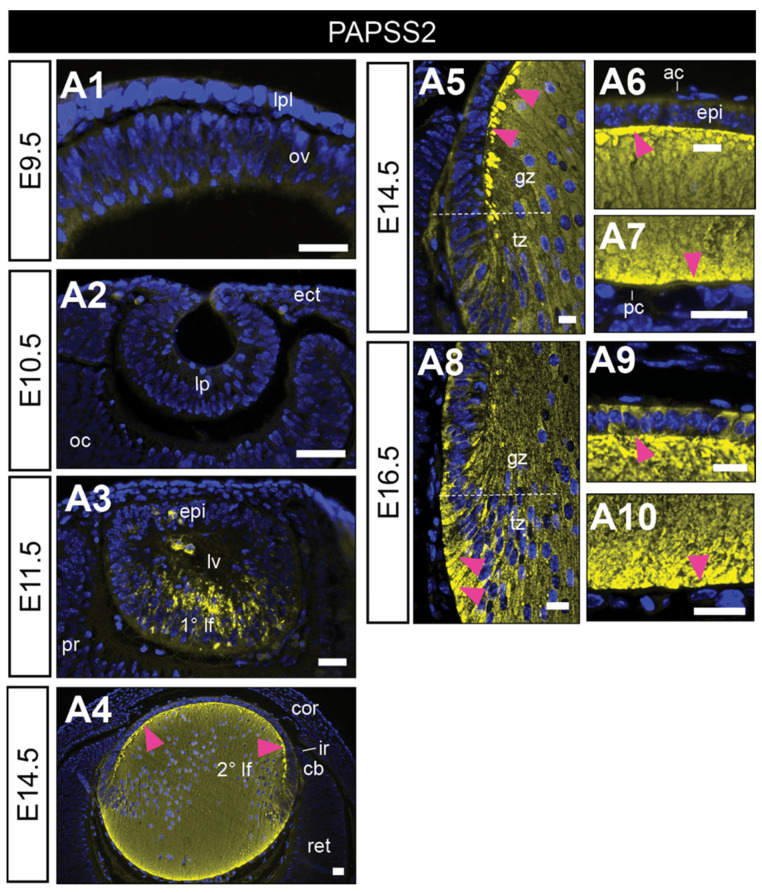
Immunolocalisation of PAPS synthetase-2 in the developing murine eye. Mid-sagittal mouse (FvB/N) eye sections from embryonic day-9.5 (E9.5; (**A1**), E10.5 (**A2**), E11.5 (**A3**), E14.5 (**A4**–**A7**) and E16.5 (**A8**–**A10**)) labelled for 3′-Phosphoadenosine 5′-Phosphosulphate Synthase 2 (PAPSS2; yellow). Nuclei counterstained with Hoechst (blue). Dotted white lines indicate lens equator (**A5**,**A8**). Magenta arrowheads indicate localisation at lens fibre tips (**A5**–**A10**). Lens placode (lpl), optic vesicle (ov), head ectoderm (ect), lens pit (lp), optic cup (oc), lens vesicle (lv), anterior lens epithelium (epi), primary lens fibre cells (1° lf), secondary lens fibre cells (2° lf), cornea (cor), retina (ret), ciliary body (cb), iris (ir), germinative zone (gz), transitional zone (tz), anterior lens capsule (ac), and posterior lens capsule (pc). Scale bars = 25 µm. Images in figures are oriented with ventral side positioned to the right and the dorsal side to the left.

**Figure 3 cells-12-01364-f003:**
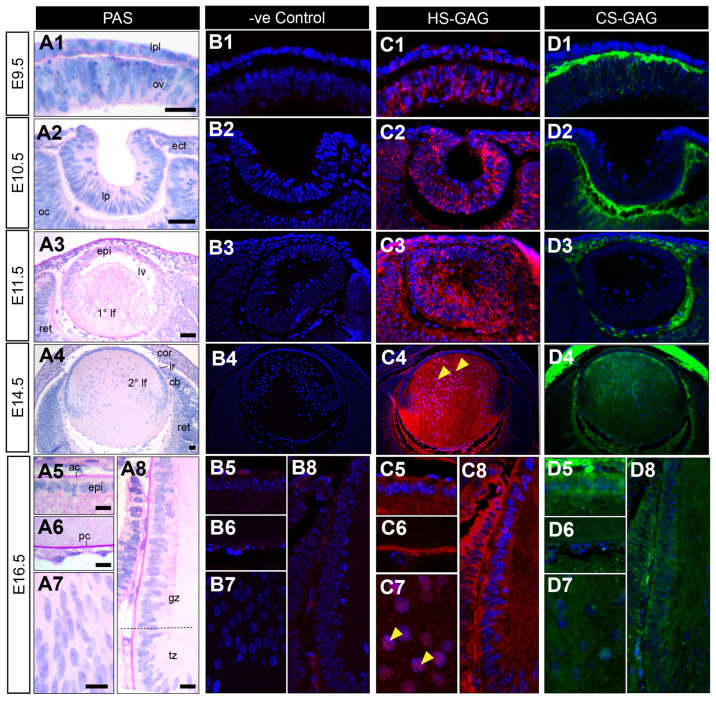
Distribution of sulphated GAGs in mouse eye development. Eye sections from embryonic day 9.5 (E9.5; (**A1**,**B1**,**C1**,**D1**), E10.5 (**A2**,**B2**,**C2**,**D2**), E11.5 (**A3**,**B3**,**C3**,**D3**), E14.5 (**A4**,**B4**,**C4**,**D4**), and E16.5 (**A5**–**A8**,**B5**–**B8**,**C5**–**C8**,**D5**–**D8**)) FvB/N mice. Mid-sagittal eye sections stained with periodic acid–Schiff’s (PAS, (**A1**–**A8**)) show cell morphology. Sections stained with a secondary antibody only show representative negative controls (negative control; (**B1**–**B8**)). Sections labelled the highly sulphated “active” forms of heparan sulphate (HS-GAG, red, (**C1**–**C8**)) or chondroitin sulphate (CS-GAG, green, (**D1**–**D8**)) glycosaminoglycan chains. Nuclei counterstained with Hoechst (blue). Yellow arrowheads indicate nuclear localisation (**C4**,**C7**). Dotted lines indicate the lens equator (**A8**). The lens placode (lpl), head ectoderm (ect), preplacodal matrix (ppm); optic vesicle (ov), lens pit (lp), optic cup (oc), lens vesicle (lv), anterior lens epithelium (epi), primary lens fibre cells (1° lf), secondary lens fibre cells (2° lf), cornea (cor), retina (ret), ciliary body (cb), iris (ir), germinative zone (gz), transitional zone (tz), anterior lens capsule (ac), posterior lens capsule (pc). Scale bars = 25 µm. Images in the figures are oriented with a ventral side positioned to the right and the dorsal side to the left.

**Figure 4 cells-12-01364-f004:**
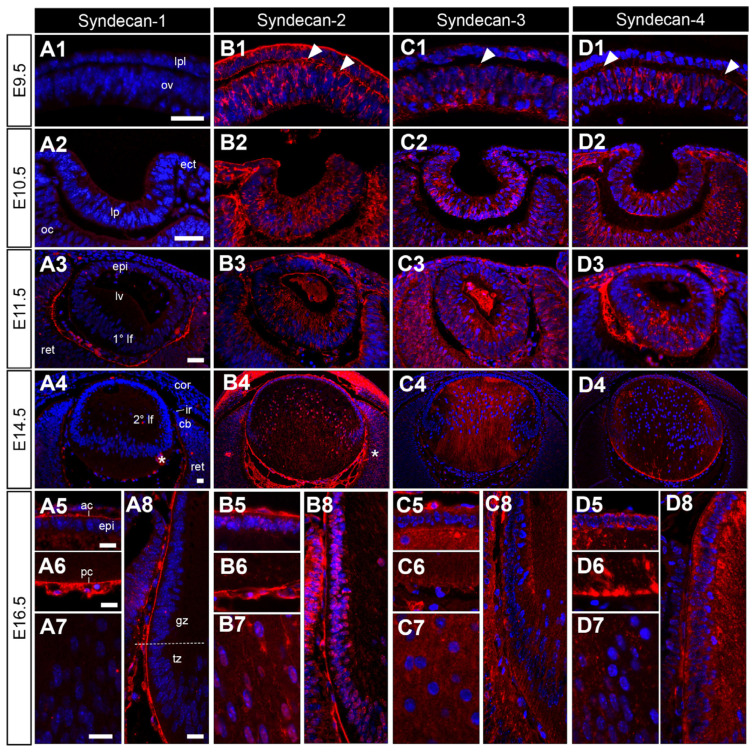
Distribution of syndecan core proteins in the developing mouse eye. Mid-sagittal sections from embryonic day-9.5 (E9.5; (**A1**,**B1**,**C1**,**D1)**), E10.5 (**A2**,**B2**,**C2**,**D2**), i5cat (**A3**,**B3**,**C3**,**D3**), E14.5 (**A4**,**B4**,**C4**,**D4**) and E16.5 (**A5**–**A8**,**B5**–**B8**,**C5**–**C8**,**D5**–**D8**)) FvB/N mice. Sections labelled for syndecan-1 (**A1**–**A8**), syndecan-2 (**B1**–**B8**), syndecan-3 (**C1**–**C8**), or syndecan-4 (**D1**–**D8**) HSPG core proteins (red). Nuclei counterstained with Hoechst (blue). Labelling of cell nuclei indicated in (**B7**,**B8**). White arrowheads indicate the staining of lens fibre cell tips (**C4**–**C6**,**C8**,**D3**, **D4**,**D6**–**D8**). The dotted line indicates the lens equator (**A8**). Lens placode (lpl), head ectoderm (ect), preplacodal matrix (ppm), optic vesicle (ov), lens pit (lp), optic cup (oc), lens vesicle (lv), anterior lens epithelium (epi), primary lens fibre cells (1° lf), secondary lens fibre cells (2° lf), cornea (cor), retina (ret), ciliary body (cb), iris (ir), germinative zone (gz), transitional zone (tz), anterior lens capsule (ac), and posterior lens capsule (pc). Scale bars = 25 µm. * denotes artefact. Images in the figures are oriented with the ventral side positioned to the right and the dorsal side to the left.

**Figure 5 cells-12-01364-f005:**
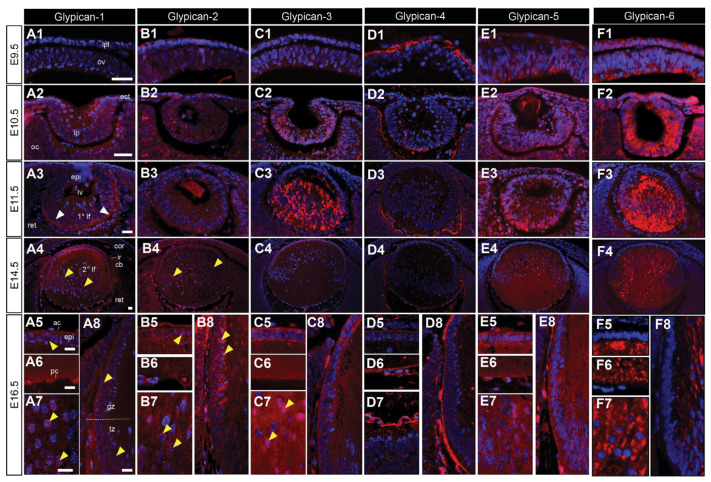
Distribution of glypican core proteins in the developing mouse eye. Mid-sagittal sections from embryonic day-9.5 (E9.5; (**A1**–**F1)**), E10.5 (**A2**–**F2**), E11.5 (**A3**–**F3**), E14.5 (**A4**–**F4**), and E16.5 (**A5**–**A8**,**B5**–**B8**,**C5**–**C8**,**D5**–**D8**,**E5**–**E8**,**F5**–**F8**) FvB/N mice. Sections labelled for glypican-1 (**A1**–**A8**), glypican-2 (**B1**–**B8**); glypican-3 (**C1–C8**), glypican-4 (**D1**–**D8**), glypican-5 (**E1**–**E8**), or glypican-6 (**F1**–**F8**) HSPG core proteins (red). Nuclei counterstained with Hoechst dye (blue). Yellow arrowheads indicate nuclear localisation (**A4**,**A5**,**A7**,**A8**,**B4**,**B5**,**B7**,**B8**,**C7**). White arrowheads indicate the staining of lens fibre cell tips (**A3**). Dotted lines indicate lens equator (**A8**). Lens placode (lpl), head ectoderm (ect), preplacodal matrix (ppm), optic vesicle (ov), lens pit (lp), optic cup (oc), lens vesicle (lv), anterior lens epithelium (epi), primary lens fibre cells (1° lf), secondary lens fibre cells (2° lf), cornea (cor), retina (ret), ciliary body (cb), iris (ir), germinative zone (gz), transitional zone (tz), anterior lens capsule (ac), and posterior lens capsule (pc). Scale bars = 25 µm. Images in figures are oriented with ventral side positioned to the right and the dorsal side to the left.

**Figure 6 cells-12-01364-f006:**
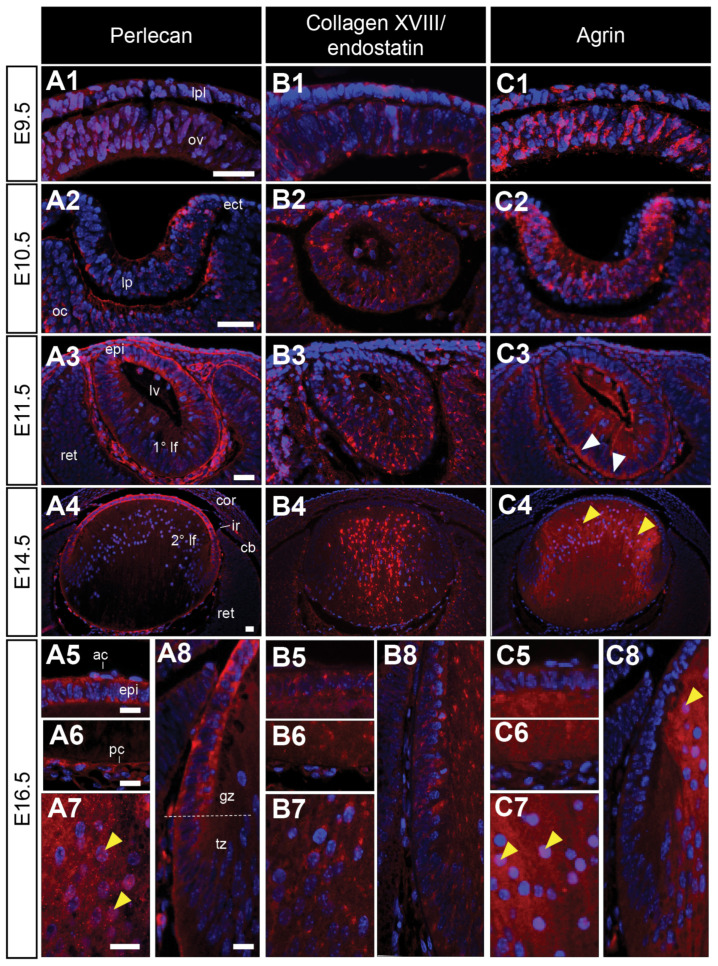
Distribution of high-molecular-weight HSPG core proteins in the developing mouse eye. Mid-sagittal sections from embryonic day-9.5 (E9.5; (**A1**–**C1)**), E10.5 (**A2**–**C2**), E11.5 (**A3**–**C3**), E14.5 (**A4**–**C4**), and E16.5 (**A5**–**A8**,**B5**–**B8**,**C5**–**C8**) FvB/N mice. Sections labelled for perlecan (**A1**–**A8**), collagen-XVIII/endostatin (**B1**–**B8**), or agrin (**C1**–**C8**) HSPG core proteins (red). Nuclei counterstained with Hoechst dye (blue). Yellow arrowheads indicate nuclear localisation (**A7**,**C4**,**C7**,**C8**). White arrowheads indicate the staining of the lens fibre cell tips (**C3**). Dotted lines indicate the lens equator (**A8**). The lens placode (lpl), head ectoderm (ect), optic vesicle (ov), lens pit (lp), optic cup (oc), lens vesicle (lv), anterior lens epithelium (epi), primary lens fibre cells (1° lf), secondary lens fibre cells (2° lf), cornea (cor), retina (ret), ciliary body (cb), iris (ir), germinative zone (gz), transitional zone (tz), anterior lens capsule (ac), and posterior lens capsule (pc). Scale bars = 25 µm. Images in figures are oriented with ventral side positioned to the right and the dorsal side to the left.

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
