# Peer review of "Spatiotemporal Localisation of Heparan Sulphate Proteoglycans throughout Mouse Lens Morphogenesis"

_cells, 2023, doi:10.3390/cells12101364_

Round 1
Reviewer 1 Report
The authors findings support a requirement for several HS-GAG sulfation in growth factor-induced lens cellular processes during embryogenesis. This paper presents important findings in the development of the lens, particularly in the process of filling the interior of the lens after the base of the capsule is formed.
In several experimental methods, important points are not described or are ambiguous in their description. The authors expect the reviewers' comments to be taken seriously.
Major points
1) As the authors note in the text from Line 434, the most careful work in this paper is to clearly position the dorsal and ventral sides of the lens during the developmental stages of lens tissue preparation. There are well-known reports of different gene and protein expression on the dorsal and ventral sides of newts during lens development. (Hayashi T. et al., Differentiation. 2002 May;70(2-3):101-8. doi: 10.1046/j.1432-0436.2002.700205.x. PMID: 12076337.) Do the authors mark the dorsal and ventral positions of the lens at the site of lens formation, or in the lens that has formed, prior to removal? (e.g., using pioctanine, etc.) If you have already positioned the specimen dorsally or ventrally before preparing the specimen, add it to the text. This may explain the difference in expression between the left and right sides in the E10.5 and E11.5 tissue specimens, as represented in Figures 3C2 and C3. If the dorsal and ventral positioning is ambiguous, this paper will not be ACCEPTED without a re-experiment.
2) Glypican-3 is found to accumulate in syncytiotrophoblasts, while glypican-1 is expressed in duodenal tissue. Glypican-1h is not expressed at E9.5 in the cornea, although the site of expression is similar to that of agrin; rather, at E9.5, Glypican-3 expression is similar to that of agrin. Since this finding is a feature of the region during development, we suggest adding a merge photo with Agrin. Add that the agrin pathway has been reported to be regulated by LIF, which plays an important role in mouse ES and iPS cells. Then cite and explain the following paper. (Terakawa J. et al., J Reprod Dev. 2009 Jun;55(3):293-8. doi: 10.1262/jrd.20162. Epub 2009 Mar 26. PMID: 19325217.)
3) Why did you use rats rather than mice as Supply data for postnatal lens data?
Minor points
1) It is unclear how many individuals are specifically used and observed at each stage. Specify the number of individual mice (crystallin lenses) used.
2) Line 408-419: The same information is repeated. Reorganize and rewrite the text.
Author Response
The authors findings support a requirement for several HS-GAG sulfation in growth factor-induced lens cellular processes during embryogenesis. This paper presents important findings in the development of the lens, particularly in the process of filling the interior of the lens after the base of the capsule is formed.
We thank the reviewer for acknowledging our important findings in relation to lens development.
In several experimental methods, important points are not described or are ambiguous in their description. The authors expect the reviewers' comments to be taken seriously.
We have addressed these as follows and assure you that we have taken the comments of the reviewer very seriously.
Major points
- As the authors note in the text from Line 434, the most careful work in this paper is to clearly position the dorsal and ventral sides of the lens during the developmental stages of lens tissue preparation. There are well-known reports of different gene and protein expression on the dorsal and ventral sides of newts during lens development. (Hayashi T. et al., Differentiation. 2002 May;70(2-3):101-8. doi: 10.1046/j.1432-0436.2002.700205.x. PMID: 12076337.) Do the authors mark the dorsal and ventral positions of the lens at the site of lens formation, or in the lens that has formed, prior to removal? (e.g., using pioctanine, etc.) If you have already positioned the specimen dorsally or ventrally before preparing the specimen, add it to the text. This may explain the difference in expression between the left and right sides in the E10.5 and E11.5 tissue specimens, as represented in Figures 3C2 and C3. If the dorsal and ventral positioning is ambiguous, this paper will not be ACCEPTED without a re-experiment.
We did not dissect and isolate the individual eyes of embryos for tissue processing. All embryonic lens tissues at all stages remained intact in the head of the embryo. We embedded, oriented and sectioned all the embryonic heads in the same manner, and as a result the ventral-dorsal orientation was maintained. While we did not make a point of any major differences in HSPG staining across the lens/eye, we have now added to the methods and figure legends that all images taken represent mid-sagittal sections of the developing eye with the ventral side to the left and the dorsal side to the right of each panel. There was no need to physically mark the tissues we collected as the surrounding head structure tissues such as the ventral nasal cavity, for example, ensure we always know the orientation.
With regards to Figures 3C2 and 3C3, while we note some very subtle changes in ventral vs dorsal labelling, this varied slightly between samples. As it is difficult to quantify the levels of labelling, we did not make a major point of ventral-dorsal differences, if any. We do acknowledge the point made by the reviewer and will allow our interested readership to come to their own conclusions.
To comment specifically on the reviewers point. In the newt, there is a great importance on dorsal vs ventral sides, especially in the regenerating lens of the newt after lensectomy as referred to in the manuscript cited by the reviewer (Hayashi et al., 2002). In contrast, our study shows the normally developing eye lens of the embryonic mouse that has not been manipulated experimentally in any way and we have maintained its orientation throughout.
- Glypican-3 is found to accumulate in syncytiotrophoblasts, while glypican-1 is expressed in duodenal tissue. Glypican-1h is not expressed at E9.5 in the cornea, although the site of expression is similar to that of agrin; rather, at E9.5, Glypican-3 expression is similar to that of agrin. Since this finding is a feature of the region during development, we suggest adding a merge photo with Agrin. Add that the agrin pathway has been reported to be regulated by LIF, which plays an important role in mouse ES and iPS cells. Then cite and explain the following paper. (Terakawa J. et al., J Reprod Dev. 2009 Jun;55(3):293-8. doi: 10.1262/jrd.20162. Epub 2009 Mar 26. PMID: 19325217.)
We thank the reviewer for this comment. At E9.5 we do not have a lens, let alone a cornea. What the reviewer is referring to is the early lens placode, a thickening of the head epidermal ectoderm. We have reviewed the images and have examined the raw data to see how close the agrin and glypican label in this region is, and we do not see any similarities. In fact, in our primary submission, we mistakenly included the exact same image for glypican 3 (5C1) and agrin (6C1) and we have now corrected this accordingly in the Figure and text. We do not see labelling for glypican 3 in the lens placode, unlike agrin, so there is no need to include a merged image of the two in the results.
- Why did you use rats rather than mice as Supply data for postnatal lens data?
We appreciate that the use of rat tissues in this study may have been confusing if the reviewer was not aware of our reference to our earlier publication (Wishart et al. 2021, IOVS). This earlier publication from our group initially examined the distribution of all HSPGs in the postnatal rat lens, a model used for decades for our experimental work. In this earlier study, while transcriptome data suggested all HSPGs were localised to the lens, we could not detect protein for glypican 5. With this current study exploring lens development in the mouse (a more pertinent developmental model), we obtained a new antibody for glypican 5 that worked well and specific showing its presence in the developing lens. To confirm glypican 5 was indeed also present in the postnatal lens, and to be consistent with our earlier manuscript, we tested it in the postnatal rat and mouse lens and reported the findings here as a Supplementary figure.
As our earlier study was over two years ago, we simply cannot add new data to this published manuscript. In fact we checked with the Editor of this journal and they are not in a position to accept any further data. For completeness and to keep the field as informed and up to date, we believe this current manuscript is the most ideal and only place we can report this finding. In short, the supplementary figure we include serves to add to the findings of the previous work/manuscript, and to provide context for the work here. We have investigated the localisation of all HSPGs including glypican-5 postnatally in both Wistar rats and FvB/N mice and found that the expression is identical. We now also make this point.
Minor points
- It is unclear how many individuals are specifically used and observed at each stage. Specify the number of individual mice (crystallin lenses) used.
We thank the reviewer for this oversight, and we have now clarified this in the Methods section.
- Line 408-419: The same information is repeated. Reorganize and rewrite the text.
We thank the reviewer for noting this, and we have now revised the text accordingly.
Reviewer 2 Report
In this manuscript, Wishart and Lovicu describe the spatio-temporal localization of 17 different GAGs, PAPSS2 and HSPG core proteins in the developing mouse lens. Overall, the writing is clear, and the microscopy images are crisp. I had only two minor comments/corrections:
1. In the introduction, the authors introduce the HSPGs abbreviation "heparan sulfate proteoglycans (HSPGs)" on line 62. They then use the abbreviation on line 65 "HSPGs (heparan sulfate proteoglycans)" defining it a second time. This is unnecessary, and the second definition should be deleted.
2. In the results, the authors abruptly shift to describing glypican-5 labaleing in the postnatal rat eye on line 285. This is described as "revised labelling" without any clear reference to the original labeling until one reads the discussion later. I think supplementary figure 1 and this description could be more appropriately sent to IOVS as a correction, so that readers of the previous paper by this group (reference 35) would be directly pointed to the corrected data. The way it is presented here would require any reader of the original 2021 IOVS paper to know that a correction has been published 2 years later in a different journal.
Author Response
In this manuscript, Wishart and Lovicu describe the spatio-temporal localization of 17 different GAGs, PAPSS2 and HSPG core proteins in the developing mouse lens. Overall, the writing is clear, and the microscopy images are crisp. I had only two minor comments/corrections:
- In the introduction, the authors introduce the HSPGs abbreviation "heparan sulfate proteoglycans (HSPGs)" on line 62. They then use the abbreviation on line 65 "HSPGs (heparan sulfate proteoglycans)" defining it a second time. This is unnecessary, and the second definition should be deleted.
We thank the reviewer for noting this, and we have now revised the text accordingly.
- In the results, the authors abruptly shift to describing glypican-5 labaleing in the postnatal rat eye on line 285. This is described as "revised labelling" without any clear reference to the original labeling until one reads the discussion later. I think supplementary figure 1 and this description could be more appropriately sent to IOVS as a correction, so that readers of the previous paper by this group (reference 35) would be directly pointed to the corrected data. The way it is presented here would require any reader of the original 2021 IOVS paper to know that a correction has been published 2 years later in a different journal.
While we acknowledge the sentiments of the reviewer, we agree with many, and have addressed it as follows. IOVS do not see this as a correction but as an addendum, so are not willing to accept further data to add to an earlier published manuscript. We have discussed this with the Editorial board member and they agreed that it is best included in the current manuscript, being sure to clearly cite the IOVS manuscript as we have done so accordingly. We would like to think that anyone reviewing this field would have access to both our earlier IOVS paper and this current paper (if it is to be published), and will make an educated and informed decision on the complete labelling for all HSPGs in embryonic and postnatal lens. As a result, we are keen to have this current body of work published as soon as possible.
In short, the supplementary figure we include serves to add to the findings of the previous work, and to provide context for the work here. We have investigated the localisation of all HSPGs including glypican-5 postnatally in both Wistar rats and FvB/N mice and found that the expression is identical. We now also make this point.
Round 2
Reviewer 1 Report
The reviewer appreciate the authors' diligent and careful revisions.